# On the Contaminated Weighted Exponential Distribution: Applications to Modeling Insurance Claim Data

Abbas Mahdavi [1], Omid Kharazmi [1] and Javier E. Contreras-Reyes [2,*]

1  Department of Statistics, Vali-e-Asr University of Rafsanjan, Rafsanjan 7718897111, Iran
2  Instituto de Estadística, Facultad de Ciencias, Universidad de Valparaíso, Valparaíso 2360102, Chile
*  Correspondence: jecontrr@uc.cl; Tel.: +56(32)-250-8242

**Abstract:** Deriving loss distribution from insurance data is a challenging task, as loss distribution is strongly skewed with heavy tails with some levels of outliers. This paper extends the weighted exponential (WE) family to the contaminated WE (CWE) family, which offers many flexible features, including bimodality and a wide range of skewness and kurtosis. We adopt Expectation-Maximization (EM) and Bayesian approaches to estimate the model, providing the likelihood and the priors for all unknown parameters. Finally, two sets of claims data are analyzed to illustrate the efficiency of the proposed method in detecting outliers.

**Keywords:** bayesian estimation; EM algorithm; Gibbs sampler; Mixture model; insurance claim data

## 1. Introduction

In many applied areas, particularly in finance and actuarial sciences, data are usually positive, right-skewed, leptokurtic and multimodal (Cummins et al. 1990). To capture a wide range of population heterogeneity and tail behavior, one practical way is to conduct analyses over subsets of claims with distinct claim characteristics. But the approach falls short of providing a full picture of claim dynamics. Classical distributions are not flexible enough to cater to heavy-tailed datasets due to extreme values that are far from the other observed data points. These unusual observations are usually called outliers. The presence of outliers in the data may distort both the estimated model parameters and the model's goodness-of-fit. Recently, many authors have focused on a finite mixture approach that shares the efficiency of parametric modeling and the flexibility of non-parametric density estimation techniques. The flexibility of finite mixtures is accommodating various shapes of insurance and economic data (Bernardi et al. 2012; Hennig and Liao 2013; Maruotti et al. 2016; Punzo et al. 2018).

Okhli and Nooghabi (2021) introduced the contaminated exponential (CE) distribution as an alternative platform for analyzing positive-valued insurance datasets with some level of outliers. The pdf of CE distribution with scale parameter $\lambda$ and contamination factor $\theta$ is defined as follows:

$$f_{CE}(y; \lambda, \theta, \omega) = (1 - \omega)\lambda e^{-\lambda y} + \omega\lambda\theta e^{-\lambda\theta y}, \quad y > 0, \lambda > 0, \tag{1}$$

where $\omega \in (0, 1)$ is the proportion of contaminated points. The Bayesian approach is developed for computing the parameter estimates. It is demonstrated that the effect of outliers is automatically reflected in the posterior distribution for any sample size. This way, an outlier observation has the highest posterior probability of outlying, but the main observations have a relatively small such probability, indicating that the CE model can detect outliers well.

Weighted distributions are used to adjust the probabilities of events as observed and recorded (Chung and Kim 2004; Gupta and Kirmani 1990; Larose and Dey 1996); (Navarro et al. 2006). Patil (1991) proceeded from applications involving statistical ecology to generate and review many useful general results concerning weighted distributions. Mild outliers, on which this paper focuses, can be dealt with by using heavy-tailed distributions

for data. Weighted distributions offer the flexibility needed for achieving mild outlier robustness, while the usual distributions like exponential, gamma and Weibull models lack sufficient fit. For more information and applications of weighted distributions see Patil and Rao (1977).

A two-parameter weighted exponential (WE) distribution (Gupta and Kundu 2009) was developed as a lifetime model which has been widely used in engineering, medicine and insurance. The sensitive skewness parameter governs essentially the shape of the probability density function (pdf) of the WE distribution. A random variable $Y$ is said to have a weighted exponential distribution with a shape parameter $\alpha > 0$, and scale parameter $\lambda > 0$, denoted by $WE(\alpha, \lambda)$, if its pdf is given by

$$f_{WE}(y; \alpha, \lambda) = \left(1 + \frac{1}{\alpha}\right) \lambda e^{-\lambda y}(1 - e^{-\alpha \lambda y}), \quad y > 0. \tag{2}$$

In this paper, we introduce a class of contaminated weighted exponential (CWE) distributions to account for all possible features of insurance and economic data. Crucially, the CWE model is a two-component mixture in which one component, with a large prior probability, represents the reference distribution, and another, with small prior probability and inflated variability, represents the degree of contamination. For Bayesian inference, we consider several asymmetric and symmetric loss functions like squared error loss, modified squared error, precautionary, weighted squared error, linear exponential, general entropy, and $K$-loss functions to estimate the parameters of the CWE model. Further, using the independent prior distributions, Bayesian 95% credible and highest posterior density (HPD) intervals (see Chen et al. 1999) are provided for each parameter of the proposed model.

The paper is organized as follows. Section 2 presents the CWE model and some illustrations of the density, skewness and kurtosis. In Sections 3 and 4, the EM algorithm and Bayesian inference are respectively developed for CWE parameters. Section 5 illustrates several simulations of proposed estimation methods of Sections 3 and 4. Sections 6 and 7 illustrates numerical examples for insurance data fitting using proposed estimation methods of Sections 3 and 4, respectively. Finally, discussions and conclusions are presented in Section 8.

## 2. The CWE Model

The pdf of a CWE model with contamination factor $\theta$ can be written as

$$f_{CWE}(y; \alpha, \lambda, \theta, \omega) = (1 - \omega)f_{WE}(y; \alpha, \lambda) + \omega f_{WE}(y; \alpha, \lambda \theta), \tag{3}$$

where $\theta > 0$ and $\omega \in [0, 1]$ denotes the proportion of outliers or unusual points and $\Theta = (\omega, \alpha, \lambda, \theta)^\top$ contains all model parameters. The CE model given in (1) is obtained as a special case of (3) when $\alpha \to \infty$. The effect of varying each parameter when one varies, but keeping others fixed, is illustrated by a set of CWE densities shown in Figure 1. The plots show that the distribution is more likely to be bimodal as $\omega$ increases, whereas flatness parameter vector $\alpha$ controls tail behavior. This implies that the CWE model provides a component of the WE distribution to capture the vast majority of small losses, whereas the contaminated component accommodates clusters of larger losses with an enhanced tail to capture extreme losses. Furthermore, the skewness and kurtosis 3D plots of the CWE model for numerous values of $\alpha$ and $\theta$ with fixed $\lambda = 1$ are depicted in the Figure 2. The fitting of this four-parameter CWE model via the likelihood approach is difficult because of the log-likelihood function's complexity. But the EM and Bayesian approaches can help.

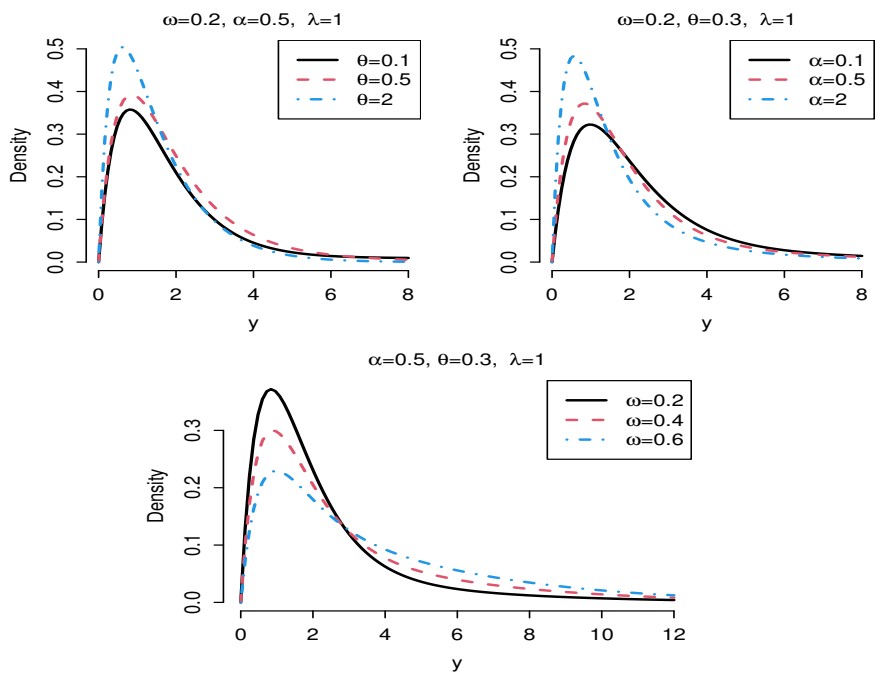

**Figure 1.** Density plots for different CWE distributions.

(a) ω=0.2

(b) ω=0.5

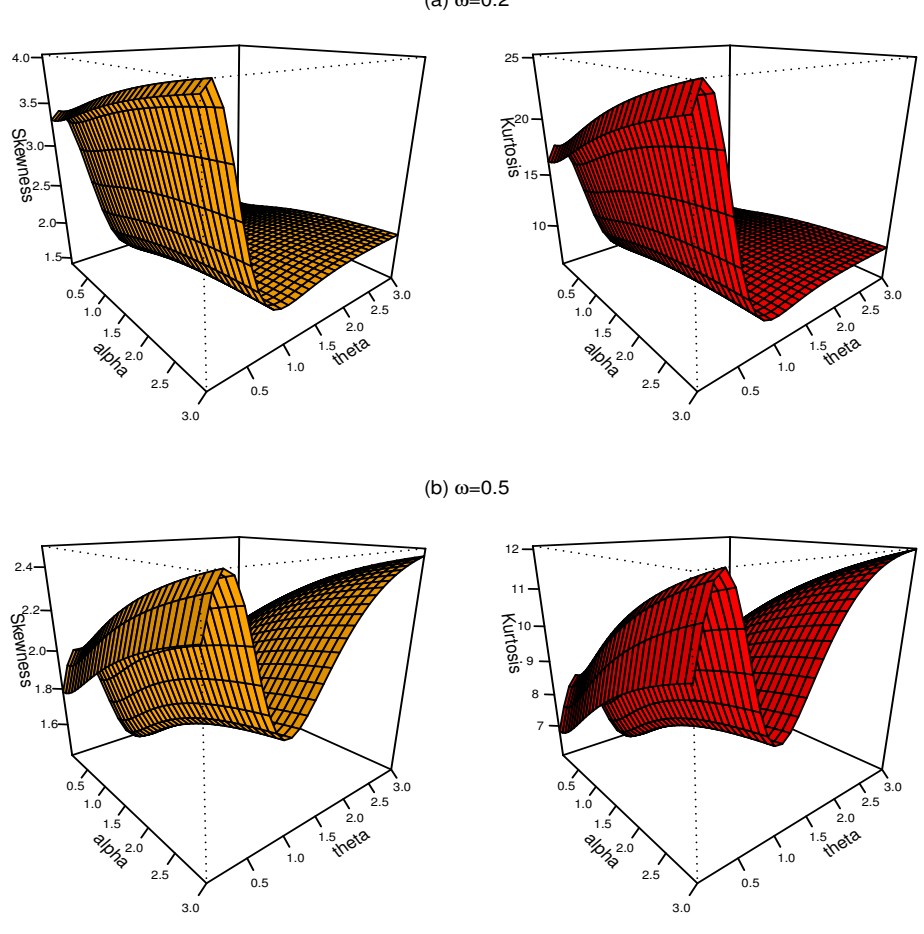

**Figure 2.** 3D plots of skewness and kurtosis of CWE distribution for two fixed values of $\omega$.

## 3. Maximum Likelihood Estimation via EM Algorithm

The EM algorithm (Dempster et al. 1977) and some of its extraordinary variants such as the expectation conditional maximization (ECM) algorithm (Meng and Rubin 1993) and the expectation-conditional maximization either (ECME) algorithm (Liu and Rubin 1994) are broadly applicable methods to carry out ML estimation for mixture distributions and variety of incomplete-data problems (Aitkin and Wilson 1980; McLachlan and Krishnan 2007; Redner and Walker 1984). Mahdavi et al. (2021a, 2021b) and Cavieres et al. (2022) developed novel EM-based procedures designed under the selection mechanism to compute the ML estimates of scale-shape mixtures of flexible generalized skew-normal and multivariate flexible skew-symmetric-normal distributions. Here, we develop a novel EM-based procedure designed under the selection mechanism to compute the ML estimates of the proposed model.

A random variable $Y \sim WE(\alpha, \lambda)$ is said to follow WE distribution with shape parameter $\alpha$ and scale parameter $\lambda$ if it has the following stochastic selection representation:

$$Y \stackrel{d}{=} X_0 | U < 1, \tag{4}$$

where $U = X_1/(\alpha X_0)$ and $X_0$ and $X_1$ are two independent exponential random variables with mean $1/\lambda$. To perform an EM-type algorithm for fitting the CWE model, we introduce a latent variable $\tau = U | U < 1$ based on (4). The joint pdf of $(Y, \tau)^\top$ is given by

$$
\begin{aligned}
f_{Y,\tau}(y, \tau) &= \frac{1}{\mathbb{P}(U < 1)} f_{X_0, U}(y, \tau) = \left(1 + \frac{1}{\alpha}\right) f_{X_0}(y) f_{U|X_0}(\tau) \\
&= (\alpha + 1)\lambda^2 y e^{-\lambda y} e^{-\lambda \alpha \tau y}, \quad y > 0, \ 0 < \tau < 1.
\end{aligned} \tag{5}
$$

Dividing (5) by (2) yields

$$f_{\tau|Y}(\tau) = \frac{\alpha \lambda y e^{-\alpha \lambda y \tau}}{1 - e^{-\alpha \lambda y}}, \quad 0 < \tau < 1. \tag{6}$$

Using (6), it is clear that

$$\tau | Y = y \sim TExp\big(\alpha \lambda y; (0, 1)\big), \tag{7}$$

where $TExp\big(\lambda; (0, b)\big)$ represents the truncated exponential distribution with mean $1/\lambda$ on interval $(0, b)$.

Let us introduce an $n$-dimensional binary random variable $\gamma = (\gamma_1, \dots, \gamma_n)^\top$ where a particular element $\gamma_i$ is equal to 1 if $Y_i$ belongs to unusual observations and is equal to zero otherwise. Note that, $\gamma_i$ follows a Bernoulli random variable with success probability $\omega$ denoted by $\gamma_i \sim Ber(\omega)$.

Now, consider $n$ independent random variables $Y_1, \dots, Y_n$, which are taken from a mixture model (3) and latent variable $\tau = (\tau_1, \dots, \tau_n)^\top$, where $\Theta = (\omega, \alpha, \lambda, \theta)^\top$ denotes the unknown vector of parameters. Clearly,

$$
\begin{aligned}
Y_i | (\gamma_i = 0) &\sim WE(\alpha, \lambda) \quad \text{and} \quad Y_i | (\gamma_i = 1) \sim WE(\alpha, \lambda \theta), \\
\tau_i | (Y_i = y_i, \gamma_i = 0) &\sim TExp(\alpha \lambda y_i; (0, 1)), \\
\tau_i | (Y_i = y_i, \gamma_i = 1) &\sim TExp(\alpha \lambda \theta y_i; (0, 1)).
\end{aligned}
$$

According to (3) and (5), it is clear that

$$f_{Y_i, \tau_i | \gamma_i}(y_i, \tau_i) = \left\{(\alpha + 1)\lambda^2 y_i e^{-\lambda y_i} e_i^{-\lambda \alpha \tau_i y_i}\right\}^{1 - \gamma_i} \left\{(\alpha + 1)\lambda^2 \theta^2 y_i e^{-\lambda \theta y_i} e^{-\lambda \theta \alpha \tau_i y_i}\right\}^{\gamma_i}.$$

The complete log-likelihood function of $\boldsymbol{\Theta}$ given $\mathbf{y}_c = (\mathbf{y}^\top, \boldsymbol{\tau}^\top, \dots, \boldsymbol{\gamma}^\top)^\top$ is

$$
\begin{aligned}
\ell_c(\boldsymbol{\Theta}|\mathbf{y}_c) &= \ln\left\{ f_{\mathbf{Y},\boldsymbol{\gamma},\boldsymbol{\tau}}(\mathbf{y},\boldsymbol{\gamma},\boldsymbol{\tau}) \right\} = \ln\left\{ f_{\boldsymbol{\gamma}}(\boldsymbol{\gamma}) f_{\mathbf{Y},\boldsymbol{\tau}|\boldsymbol{\gamma}}(\mathbf{y},\boldsymbol{\tau}) \right\} \\
&= \sum_{i=1}^n \Big\{ \gamma_i \ln \omega + (1-\gamma_i)\ln(1-\omega) + \ln(\alpha+1) + 2\ln\lambda + 2\gamma_i \ln\theta \\
&\qquad - (1-\gamma_i)\lambda y_i - (1-\gamma_i)\lambda\alpha\tau_i y_i - \gamma_i\lambda\theta y_i - \gamma_i\lambda\theta\alpha\tau_i y_i \Big\}.
\end{aligned}
\tag{8}
$$

To evaluate the *Q*-function, the necessary conditional expectations include

$$
\begin{aligned}
\hat{\gamma}_i^{(k)} &= E\big(\gamma_i|Y_i=y_i, \hat{\boldsymbol{\Theta}}^{(k)}\big) = \frac{\hat{\omega}^{(k)} f_{WE}(y_i; \hat{\alpha}^{(k)}, \hat{\lambda}^{(k)} \hat{\boldsymbol{\Theta}}^{(k)})}{f_{CWE}(y_i; \hat{\alpha}^{(k)}, \hat{\lambda}^{(k)}, \hat{\boldsymbol{\Theta}}^{(k)})}, \\
\hat{\tau}_{1i}^{(k)} &= E\big((1-\gamma_i)\tau_i|Y_i=y_i, \hat{\boldsymbol{\Theta}}^{(k)}\big) = (1-\hat{\gamma}_i^{(k)})\left( \frac{1}{\hat{\alpha}^{(k)}\hat{\lambda}^{(k)}y_i} - \frac{1}{e^{\hat{\alpha}^{(k)}\hat{\lambda}^{(k)}y_i} - 1} \right), \\
\hat{\tau}_{2i}^{(k)} &= E\big(\gamma_i\tau_i|Y_i=y_i, \hat{\boldsymbol{\Theta}}^{(k)}\big) = \hat{\gamma}_i^{(k)}\left( \frac{1}{\hat{\alpha}^{(k)}\hat{\lambda}^{(k)}\hat{\theta}^{(k)}y_i} - \frac{1}{e^{\hat{\alpha}^{(k)}\hat{\lambda}^{(k)}\hat{\theta}^{(k)}y_i} - 1} \right).
\end{aligned}
$$

Therefore, the *Q*-function is given by

$$
\begin{aligned}
Q(\boldsymbol{\Theta}|\hat{\boldsymbol{\Theta}}^{(k)}) = \sum_{i=1}^n \Big\{ &\hat{\gamma}_i^{(k)}\ln\omega + (1-\hat{\gamma}_i^{(k)})\ln(1-\omega) + \ln(\hat{\alpha}^{(k)}+1) + 2\ln\hat{\lambda}^{(k)} \\
&+ 2\hat{\gamma}_i^{(k)}\ln\hat{\theta}^{(k)} - \hat{\lambda}^{(k)}(1-\hat{\gamma}_i^{(k)})y_i - \hat{\lambda}^{(k)}\hat{\alpha}^{(k)}\hat{\tau}_{1i}^{(k)}y_i \\
&- \hat{\lambda}^{(k)}\hat{\theta}^{(k)}\hat{\gamma}_i^{(k)}y_i - \hat{\lambda}^{(k)}\hat{\theta}^{(k)}\hat{\alpha}^{(k)}\hat{\tau}_{2i}^{(k)}y_i \Big\}.
\end{aligned}
\tag{9}
$$

In summary, the implementation of the ECM algorithm proceeds as follows:

**E-step:** Given $\boldsymbol{\Theta} = \hat{\boldsymbol{\Theta}}^{(k)}$, compute $\hat{\gamma}_i^{(k)}$, $\hat{\tau}_{1i}^{(k)}$ and $\hat{\tau}_{2i}^{(k)}$ for $i = 1, \dots, n$.

**CM-step 1:** Calculate

$$
\hat{\omega}^{(k+1)} = \frac{1}{n}\sum_{i=1}^n \hat{\gamma}_i^{(k)}.
$$

**CM-step 2:** Fix $\lambda = \hat{\lambda}^{(k)}$, $\theta = \hat{\theta}^{(k)}$ and update $\hat{\alpha}^{(k)}$ by maximizing (9) over $\alpha$, which gives

$$
\hat{\alpha}^{(k+1)} = \frac{n}{\hat{\lambda}^{(k)}\sum_{i=1}^n \big(\hat{\tau}_{1i}^{(k)}y_i - \hat{\theta}^{(k)}\hat{\tau}_{2i}^{(k)}y_i\big)} - 1.
$$

**CM-step 3:** Fix $\alpha = \hat{\alpha}^{(k+1)}$, $\theta = \hat{\theta}^{(k)}$ and update $\hat{\lambda}^{(k)}$ by

$$
\hat{\lambda}^{(k+1)} = \frac{2n}{\sum_{i=1}^n \big\{ (1-\hat{\gamma}^{(k)})y_i + \hat{\alpha}^{(k+1)}\hat{\tau}_{1i}^{(k)}y_i + \hat{\theta}^{(k)}\hat{\gamma}_i^{(k)}y_i + \hat{\alpha}^{(k+1)}\hat{\theta}^{(k)}\hat{\tau}_{2i}y_i \big\}}.
$$

**CM-step 4:** Fix $\alpha = \hat{\alpha}^{(k+1)}$, $\lambda = \hat{\lambda}^{(k+1)}$ and update $\hat{\theta}^{(k)}$ by

$$
\hat{\theta}^{(k+1)} = \frac{2\sum_{i=1}^n \hat{\gamma}_i^{(k)}}{\hat{\lambda}^{(k+1)}\sum_{i=1}^n \big\{ \hat{\gamma}_i^{(k)}y_i + \hat{\alpha}^{(k+1)}\hat{\tau}_{2i}^{(k)}y_i \big\}}.
$$

This process is repeated until a suitable convergence rule is satisfied. The convergence appears when the relative difference between two successive log-likelihood values is less than tolerance ($\epsilon$). In our numerical experiments, $\epsilon = 10^{-6}$ is used. An R code about EM algorithm is available in Appendix A.

## 4. Bayesian Inference

In this section, we discuss the Bayesian estimation for the CWE distribution parameters in terms of several symmetric and asymmetric loss functions such as squared error loss function (SELF), weighted squared error loss function (WSELF), modified squared error loss function (MSELF), precautionary loss function (PLF) and K-loss function (KLF). The considered loss functions and their Bayesian estimators with corresponding posterior risks are reported in Table 1.

**Table 1.** Bayes estimator and posterior risk under several loss functions.

| Loss Function $L(\psi, \delta)$ | Bayes Estimator $\psi_B$ | Posterior Risk $\rho_\psi$ |
|---|---|---|
| $SELF = (\psi - d)^2$ | $E(\psi|x)$ | $Var(\psi|x)$ |
| $WSELF = \frac{(\psi-d)^2}{\psi}$ | $(E(\psi^{-1}|x))^{-1}$ | $E(\psi|x) - (E(\psi^{-1}|x))^{-1}$ |
| $MSELF = \left(1 - \frac{d}{\psi}\right)^2$ | $\frac{E(\psi^{-1}|x)}{E(\psi^{-2}|x)}$ | $1 - \frac{E(\psi^{-1}|x)^2}{E(\psi^{-2}|x)}$ |
| $PLF = \frac{(\psi-d)^2}{d}$ | $\sqrt{E(\psi^2|x)}$ | $2\left(\sqrt{E(\psi^2|x)} - E(\psi|x)\right)$ |
| $KLF = \left(\sqrt{\frac{d}{\psi}} - \sqrt{\frac{\psi}{d}}\right)$ | $\sqrt{\frac{E(\psi|x)}{E(\psi^{-1}|x)}}$ | $2\left(\sqrt{E(\psi|x)E(\psi^{-1}|x)} - 1\right)$ |

For pertinent details about these loss functions, refer to Kharazmi et al. (2021, 2022) and references therein.

### 4.1. Joint and Marginal Posterior Distributions

Assume that the parameters of the CWE distribution have independent prior distributions as follows: $\alpha \sim Gamma(\alpha_0, \alpha_1)$, $\theta \sim Gamma(\theta_0, \theta_1)$, $\lambda \sim Gamma(\lambda_0, \lambda_1)$, and $\omega \sim Beta(\omega_0, \omega_1)$, where all hyper-parameters are positive. Consequently, the joint prior density is formulated as

$$\pi(\alpha, \lambda, \theta, \omega) = \frac{\omega^{\omega_0}(1-\omega)^{\omega_1}\alpha_1^{\alpha_0}\theta_1^{\theta_0}\lambda_1^{\lambda_0}}{Beta(\omega_0, \omega_1)\Gamma(\alpha_0)\Gamma(\theta_0)\Gamma(\lambda_0)}\alpha^{\alpha_0-1}\theta^{\theta_0-1}\lambda^{\lambda_0}e^{-(\alpha_1\alpha+\theta_1\theta+\lambda_1\lambda)}.$$

For simplicity, we define function $\zeta$ as

$$\zeta(\alpha, \theta, \lambda, \omega) = \alpha^{\alpha_0-1}\beta^{\beta_0-1}\lambda^{\lambda_0}e^{-(\alpha_1\alpha+\beta_1\beta+\lambda_1\lambda)}\omega^{\omega_0}(1-\omega)^{\omega_1}.$$

From (10) and likelihood function $L(data)$, the joint posterior distribution is

$$\pi^*(\alpha, \theta, \lambda, \omega|data) \propto \pi(\alpha, \theta, \lambda, \omega)\, L(data).$$

Therefore, the exact joint posterior pdf is given by

$$\pi^*(\alpha, \theta, \lambda, \omega|\underline{x}) = K\zeta(\alpha, \theta, \lambda, \omega)\, L(\underline{x}, \Psi), \tag{10}$$

where

$$L(\underline{x}; \Psi) = \left[\lambda\left(1 + \frac{1}{\alpha}\right)\right]^n \prod_{i=1}^n \left\{(1-\omega)e^{-\lambda x_i}(1 - e^{-\alpha\lambda x_i}) + \omega\theta e^{-\theta\lambda x_i}(1 - e^{-\alpha\theta\lambda x_i})\right\}, \tag{11}$$

$\Psi = (\alpha, \theta, \lambda, \omega)$ and $K$ is a normalizing constant with form

$$K^{-1} = \int_0^1 \int_0^\infty \int_0^\infty \int_0^\infty \zeta(\alpha, \theta, \lambda, \omega)L(\underline{x}, \xi)\partial\alpha\partial\beta\partial\lambda\partial\omega.$$

Moreover, the marginal posterior density of $\alpha$, $\theta$, $\lambda$ and $\omega$ (assuming $\Psi = (\Psi_1, \Psi_2, \Psi_3, \Psi_4) = (\alpha, \theta, \lambda, \omega)$) can be expressed as

$$\pi(\Psi_i|\underline{x}) = \begin{cases} \int_0^1 \int_0^\infty \int_0^\infty \pi^*(\Psi|\underline{x}) \partial\Psi_j \partial\Psi_k \partial\Psi_4, & i = 1,2,3, \\ \int_0^\infty \int_0^\infty \int_0^\infty \pi^*(\Psi|\underline{x}) \partial\Psi_1 \partial\Psi_2 \partial\Psi_3, & i = 4, \end{cases} \tag{12}$$

where $j, k = 1, 2, 3$, $j \neq k \neq i$ and $\Psi_i$ is the $i$th member of vector $\Psi$.

### 4.2. Bayesian Point Estimation

From the marginal posterior pdf in (12) and under framework of the loss functions listed in Table 1, the Bayesian point estimation for parameter vector $\Psi = (\Psi_1, \Psi_2, \Psi_3, \Psi_4) = (\alpha, \theta, \lambda, \omega)$ is formulated via minimizing the expectation of loss function with respect to the marginal posterior pdf in (12) as follows:

$$\operatorname{argmin} C_\delta \int_0^\infty L(\Psi_i, \delta) \pi(\Psi_i|\underline{x}) \partial\Psi_i. \tag{13}$$

In practice, because of the intractable integral in (13), we can use the Gibbs sampler (Geman and Geman 1984) or Metropolis-Hastings algorithms (Hastings 1970; Metropolis et al. 1953) to generate posterior samples. We will argue this issue more precisely in Section 4.5.

### 4.3. Credibility Interval

In the Bayesian framework, interval estimation is done via credibility interval conception. Consider parameter vector $\Psi = (\Psi_1, \Psi_2, \Psi_3, \Psi_4) = (\alpha, \theta, \lambda, \omega)$, which is associated with CWE distribution and $\pi(\Psi_j|\underline{x})$ the marginal posterior pdf of parameter $\Psi_j$, $j = 1, 2, 3, 4$, as in (12). For a given value of $\eta \in (0, 1)$, the $(1 - \eta)100\%$ credibility interval $CI(L_{\Psi_j}, U_{\Psi_j})$ is defined as

$$\int_{L_{\Psi_j}}^\infty \pi(\Psi_j|\underline{x}) \partial\Psi_j = 1 - \frac{\eta}{2}, \tag{14}$$

$$\int_{U_{\Psi_j}}^\infty \pi(\Psi_j|\underline{x}) \partial\Psi_j = \frac{\eta}{2}. \tag{15}$$

By considering relation (14) and (15), it is not feasible to obtain the explicit marginal pdf from the joint posterior distribution. To overcome this difficulty, we use the Gibbs sampler algorithm and generate posterior samples from the CWE distribution. Let $\Psi^1, \ldots, \Psi^k$ (where $\Psi^i = (\Psi_1^i, \Psi_2^i, \Psi_3^i, \Psi_4^i)$) be a posterior random sample of size $k$ which is extracted from the joint posterior pdf in (10). Using these samples, the marginal posterior pdf of $\Psi_j$ given $\underline{x}$ is defined by

$$\frac{1}{K} \sum_{i=1}^K \pi^*(\Psi_j, \Psi_{-j}^i|\underline{x}), \quad j = 1, 2, 3, 4, \tag{16}$$

where $\Psi_{-j}^i$ represents the vector of posterior samples when the $j$th component is removed. Inserting (16) in (15), it is possible to compute the credibility intervals for $\Psi_j$, $j = 1, 2, 3, 4$, as follows

$$\frac{1}{K} \sum_{i=1}^K \int_{L_{\Psi_j}}^\infty \pi^*(\Psi_j, \Psi_{-j}^i|\underline{x}) \partial\Psi_j = 1 - \frac{\eta}{2}, \tag{17}$$

$$\frac{1}{K} \sum_{i=1}^K \int_{U_{\Psi_j}}^\infty \pi^*(\Psi_j, \Psi_{-j}^i|\underline{x}) \partial\Psi_j = \frac{\eta}{2}. \tag{18}$$

*4.4. Highest Posterior Density Interval*

Highest posterior density (HPD) interval is a credibility interval under a specific restriction. A $(1 - \eta)100\%$ HPD interval for $\Psi_j$, $j = 1, 2, 3, 4$ is the simultaneous solution of integral equations

$$\frac{1}{K} \sum_{i=1}^{K} \int_{L_{\Psi_j}}^{U_{\Psi_j}} \pi^*\left(\Psi_j, \Psi_{-j}^i | \underline{x}\right) \partial \Psi_j \;\; = \;\; 1 - \eta, \tag{19}$$

$$\sum_{i=1}^{K} \pi^*\left(L_{\Psi_j}, \Psi_{-j}^i | \underline{x}\right) \;\; = \;\; \sum_{i=1}^{K} \pi^*\left(U_{\Psi_j}, \Psi_{-j}^i | \underline{x}\right). \tag{20}$$

*4.5. Generating Posterior Samples*

It is clear from Equations (10) and (12) that there are no explicit expressions for the Bayesian point estimators under the loss functions in Table 1. Because of intractable integrals associated with joint posterior and marginal posterior distributions, we require numerical software to solve the integral equations numerically via MCMC methods such as the Metropolis-Hastings algorithm and Gibbs sampling (Contreras-Reyes et al. 2018). Assuming general model $f(\underline{x}|\boldsymbol{\psi})$ is associated with parameter vector $\boldsymbol{\psi} = (\psi_1, \psi_2, \ldots, \psi_p)$ and observed data $\underline{x}$, the joint posterior distribution is $\pi(\psi_1, \psi_2, \ldots, \psi_p | \underline{x})$. We also assume that $\boldsymbol{\psi_0} = (\psi_1^{(0)}, \psi_2^{(0)}, \ldots, \psi_p^{(0)})$ is the initial vector to start the Gibbs sampler (Quintero et al. 2017). The steps for any iteration, say iteration $k$, are as follows:

- Starting with an initial estimate $(\psi_1^{(0)}, \psi_2^{(0)}, \ldots, \psi_p^{(0)})$;
- draw $\psi_1^k$ from $\pi(\psi_1 | \psi_2^{k-1}, \psi_3^{k-1}, \ldots, \psi_p^{k-1}, \underline{x})$;
- draw $\psi_2^k$ from $\pi(\psi_2 | \psi_1^k, \psi_3^{k-1}, \ldots, \psi_p^{k-1}, \underline{x})$; and so on down to
- draw $\psi_p^k$ from $\pi(\psi_p | \psi_1^k, \psi_2^k, \ldots, \psi_{p-1}^k, \underline{x})$.

In the case of the CWE distribution, by considering parameter vector $\Psi = (\alpha, \theta, \lambda, \omega)$ and initial parameter vector $\Psi_0 = (\alpha^0, \theta^0, \lambda^0, \omega^0)$, the posterior samples are extracted based on Gibbs sampler where the full conditional distributions are

$$\pi\left(\alpha | \theta^{k-1}, \lambda^{k-1}, \omega^{k-1}, \underline{x}\right) \propto \left(\frac{\alpha+1}{\alpha}\right)^n \alpha^{\alpha_0} \, e^{-\alpha_1 \alpha} \prod_{i=1}^{n} Y(x_i, \Psi), \tag{21}$$

$$\pi\left(\theta | \alpha^{k-1}, \lambda^{k-1}, \omega^{k-1}, \underline{x}\right) \propto \beta^{\theta_0} \, e^{-\theta_1 \theta} \prod_{i=1}^{n} Y(x_i, \Psi), \tag{22}$$

$$\pi\left(\lambda | \alpha^{k-1}, \theta^{k-1}, \omega^{k-1}, \underline{x}\right) \propto \lambda^{\lambda_0 + n} e^{-\lambda_1 \lambda} \prod_{i=1}^{n} Y(x_i, \Psi), \tag{23}$$

and

$$\pi\left(\omega | \alpha^{k-1}, \theta^{k-1}, \lambda^{k-1}, \underline{x}\right) \propto \omega^{\omega_0} (1 - \omega)^{\omega_1} \prod_{i=1}^{n} Y(x_i, \Psi), \tag{24}$$

where $Y(x_i, \Psi) = (1 - \omega)e^{-\lambda x_i}(1 - e^{-\alpha \lambda x_i}) + \omega \theta e^{-\theta \lambda x_i}(1 - e^{-\alpha \theta \lambda x_i})$.

In practice, simulations related to Gibbs sampling can be done with special software WinBUGS. This software was developed in 1997 to simulate data of complex posterior distributions, where analytical or numerical integration techniques cannot be applied. Moreover, Gibbs sampling processes can be carried out via OpenBUGS software, which is an open source version of WinBUGS. Since there isn't any prior information about hyper-parameters in (10), we follow Congdon (2001) and the hyper-parameter values are set as $\alpha_i = \theta_i = \lambda_i = \omega_i = 0.0001$, $i = 0, 1$, so we can use the MCMC procedure to extract posterior samples of (10) by means of Gibbs sampling process in OpenBUGS software.

## 5. Simulation Study: Recovery of the True Underlying Parameters

An experiment intends to investigate the ability of the proposed EM algorithm to recover the true underlying parameters. We generate 5000 synthetic Monte Carlo samples

of different sample sizes $n = 30, 70, 100$ and $200$ from the CWE distribution and following three parameter scenarios (each scenario corresponding to density plotted as "dotdash" line in Figure 1):

**Scenario 1:** $\alpha = 0.5, \lambda = 1, \theta = 2, \omega = 0.2$.
**Scenario 2:** $\alpha = 2, \lambda = 1, \theta = 0.3, \omega = 0.2$.
**Scenario 3:** $\alpha = 0.5, \lambda = 1, \theta = 0.3, \omega = 0.6$.

The accuracies of the parameter estimates are measured by computing the mean absolute bias (*MAB*) and the root mean square error (*RMSE*), defined as

$$MAB = \frac{1}{5000} \sum_{i=1}^{5000} |\hat{\theta}_i - \theta_A| \quad \text{and} \quad RMSE = \sqrt{\frac{1}{5000} \sum_{i=1}^{5000} (\hat{\theta}_i - \theta_A)^2},$$

where $\hat{\theta}_i$ denotes the prediction of a specific parameter at the $i$-th replication and $\theta_A$ denotes the actual specific parameter value. Table 2 shows the simulation results for the CWE distribution. As expected, the MAB and RMSE tend toward zero when the sample size increases, showing empirically the consistency of the ML estimates obtained via the EM algorithm.

**Table 2.** Simulation results, based on 5000 replications, to evaluate the EM algorithm under three scenarios.

| Sample Size | Parameter | $n = 30$ | | $n = 70$ | | $n = 100$ | | $n = 200$ | |
|---|---|---|---|---|---|---|---|---|---|
| | | MAB | RMSE | MAB | RMSE | MAB | RMSE | MAB | RMSE |
| Scenario 1 | $\alpha$ | 0.357 | 0.419 | 0.270 | 0.329 | 0.232 | 0.284 | 0.171 | 0.213 |
| | $\lambda$ | 0.204 | 0.259 | 0.139 | 0.176 | 0.118 | 0.150 | 0.083 | 0.104 |
| | $\theta$ | 1.906 | 7.158 | 1.176 | 2.943 | 0.955 | 1.825 | 0.645 | 1.003 |
| | $\omega$ | 0.094 | 0.148 | 0.073 | 0.109 | 0.065 | 0.093 | 0.050 | 0.069 |
| Scenario 2 | $\alpha$ | 1.749 | 2.892 | 1.165 | 1.626 | 0.959 | 1.286 | 0.697 | 0.898 |
| | $\lambda$ | 0.308 | 0.419 | 0.197 | 0.259 | 0.163 | 0.213 | 0.115 | 0.148 |
| | $\theta$ | 0.189 | 0.282 | 0.102 | 0.151 | 0.080 | 0.115 | 0.052 | 0.069 |
| | $\omega$ | 0.118 | 0.158 | 0.098 | 0.126 | 0.088 | 0.112 | 0.069 | 0.087 |
| Scenario 3 | $\alpha$ | 0.449 | 0.665 | 0.359 | 0.508 | 0.306 | 0.412 | 0.236 | 0.310 |
| | $\lambda$ | 0.366 | 0.564 | 0.229 | 0.323 | 0.187 | 0.252 | 0.132 | 0.172 |
| | $\theta$ | 0.092 | 0.121 | 0.059 | 0.075 | 0.050 | 0.064 | 0.036 | 0.045 |
| | $\omega$ | 0.169 | 0.204 | 0.126 | 0.155 | 0.109 | 0.135 | 0.082 | 0.102 |

## 6. Numerical Examples for Insurance Data Fitting

In this section, we evaluate the performance and various aspects of the proposed model using insurance claims data. The proposed distribution is fitted to the data by implementing the ECM algorithm described in Section 3. For the sake of comparison, the reduced WE, CE and exponential (Exp) models are also fitted as sub-models of CWE distribution. To compare how well the models fit the data, we adopt the Akaike information criterion (AIC) (Akaike 1973) and the Bayesian information criterion (BIC) (Schwarz 1978), defined as AIC $= 2p - 2\ell_{max}$ and BIC $= p \log n - 2\ell_{max}$, where $p$ is the number of free parameters in the model and $\ell_{max}$ the maximized log-likelihood value. For both AIC and BIC, a smaller value indicates a better model fit.

The first dataset (DS1) comprises Danish fire losses analyzed in McNeil (1997). This dataset is frequently used for comparison of methods; see Eling (2012) and references therein. These data represent Danish fire losses in million Danish Krones and were collected by a Danish reinsurance company. The dataset contains individual losses above 1 million Danish Krones, a total of 2167 individual losses, covering the period from 3 January 1980 to 31 December 1990. Data are adjusted for inflation to reflect 1985 values and are available in R packages `evir` and `fExtremes`.

The second dataset (DS2), analyzed by Cummins and Freifelder (1978), contains 80 fire losses from 500 buildings a large university owned from 1951 to 1973. Cummins et al.

(1990) found that the log-normal and gamma distributions did not have sufficient heavy tails to model the data, so they considered the generalized beta of the second kind (GB2) distribution.

Figure 3 presents two histograms for the considered datasets. Both histograms reveal a typical feature of insurance claims data: a large number of small losses and a small number of very large losses. Table 3 reports parameter estimates, standard error and model fit criteria for all fitted models. Observing the Table 3, it is evident from the AIC and BIC values that the CWE model provides better fit than other fitted models. The posterior probability of each observation belonging to unusual observations is depicted in Figures 4 and 5, those reveal that the unusual data have the highest posterior probability and the original data have small posterior probability, showing clearly the impact of outliers.

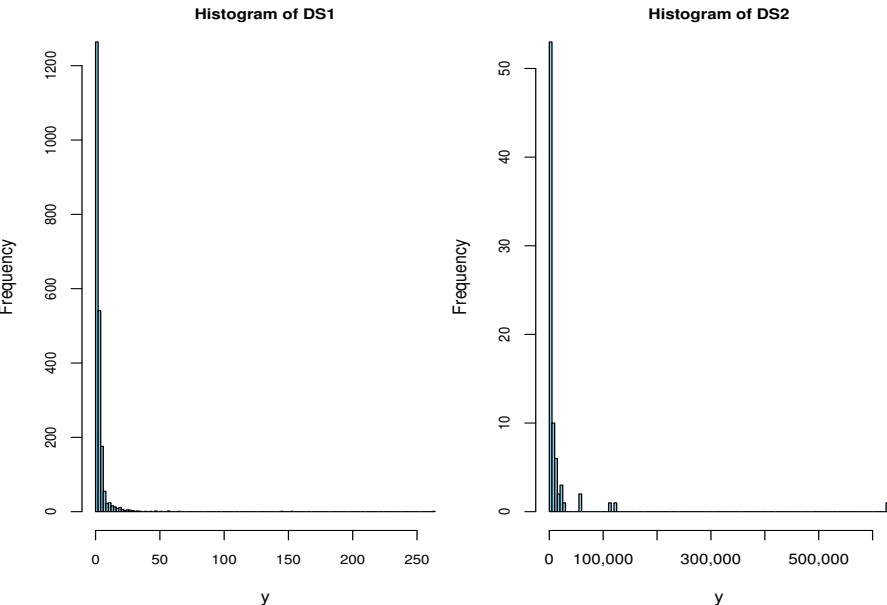

**Figure 3.** Data histograms corresponding to DS1 and DS2 datasetes.

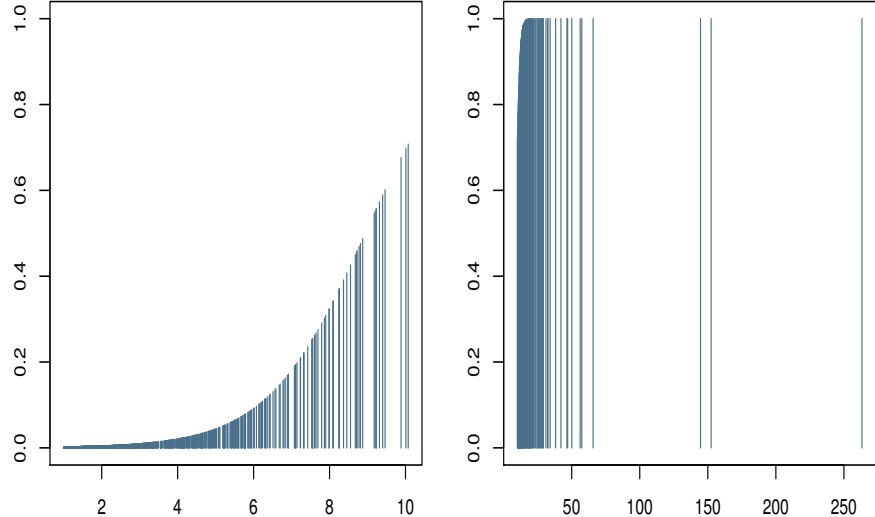

**Figure 4.** Posterior probability that each observation is unusual, corresponding to DS1 dataset. (**Left**) panel is for the first 2060 observations and (**right**) panel for the 107 last observations.

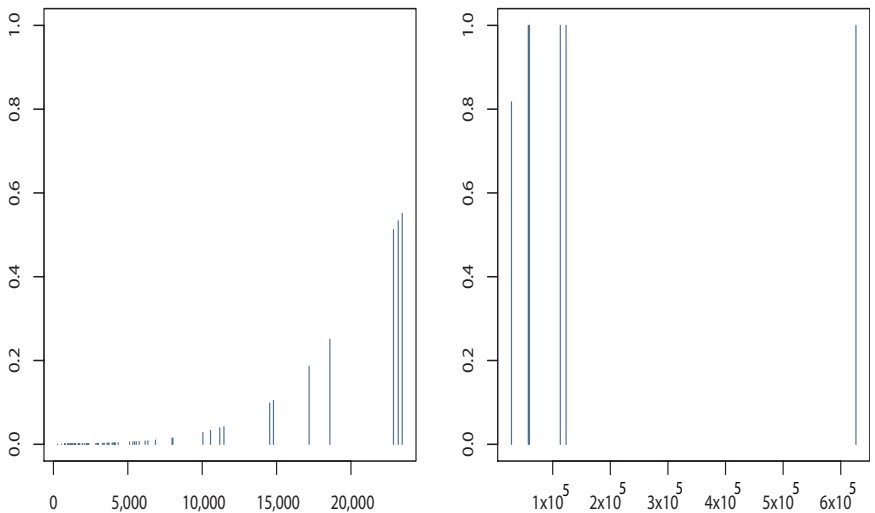

**Figure 5.** Posterior probability that each observation is unusual, corresponding to the DS2 dataset. Left panel is for the first 74 observations and right panel for the six last observations.

**Table 3.** Summary results from fitting various models to the data. The bold entries highlight the smallest AIC and BIC values for each model.

| Dataset | Model | $\hat{\lambda}$ | $\hat{\theta}$ | $\hat{\alpha}$ | $\hat{\omega}$ | $p$ | $\ell_{max}$ | AIC | BIC |
|---------|-------|-----------------|----------------|----------------|----------------|-----|--------------|-----|-----|
| DS1 | Exp | 0.295 | – | – | – | 1 | −4809.396 | 9620.792 | 9626.474 |
| | WE | 0.350 | – | 4.420 | – | 2 | −4576.327 | 9160.655 | 9183.379 |
| | CE | 0.401 | 0.107 | – | 0.043 | 3 | −4556.646 | 9119.292 | 9136.335 |
| | CWE | 0.818 | 0.113 | 0.194 | 0.064 | 4 | **−4119.475** | **8246.950** | **8269.675** |
| DS2 | Exp | $0.590 \times 10^{-5}$ | – | – | – | 1 | −859.0414 | 1720.083 | 1722.465 |
| | WE | $0.596 \times 10^{-4}$ | – | 103.667 | – | 2 | −858.548 | 1725.096 | 1734.624 |
| | CE | $0.223 \times 10^{-3}$ | 0.033 | – | 0.096 | 3 | −796.815 | 1599.630 | 1606.776 |
| | CWE | $0.258 \times 10^{-3}$ | 0.035 | 9.112 | 0.104 | 4 | **−793.087** | **1594.175** | **1603.703** |

## 7. Bayesian Numerical Results

We used an MCMC procedure based on 10,000 replicates with 1000 samples discarded as burn-in to compute the Bayesian estimators. The corresponding Bayesian point estimation and posterior risk based on DS1 and DS2 datasets are provided in Table 4. It can be seen that for the both datasets, the resulting log-likelihood values ($\ell_{max}$) are close to the obtained ones by the EM-algorithm given in Table 3, indicating the efficiency of the Bayesian approach to estimate the model parameters. It is noteworthy to mention that the KLF and PLF loss functions yields the highest log-likelihood values for DS1 and DS2 datasets, respectively.

Table 5 provides 95% credible and HPD intervals for the parameters of the CWE distribution. The posterior samples are extracted using Gibbs sampling technique. Moreover, we provide the posterior summary plots in Figures 6–8. These plots confirm that the convergence of the Gibbs sampling process occurred.

**Table 4.** Bayesian estimates and their posterior risks of the CWE distribution parameters under different loss functions based on DS1 and DS2 datasets. The bold entries highlight the highest $\ell_{max}$ values for each model.

| Data | DS1 | | | | |
|---|---|---|---|---|---|
| **Bayesian Estimation** | | | | | |
| **Loss Function** | $\widehat{\lambda}$ ($r_{\widehat{\lambda}}$) | $\widehat{\theta}$ ($r_{\widehat{\theta}}$) | $\widehat{\alpha}$ ($r_{\widehat{\alpha}}$) | $\widehat{\omega}$ ($r_{\widehat{\omega}}$) | $\ell_{max}$ |
| SELF | 0.74215 (0.00119) | 0.10790 (0.00006) | 0.48874 (0.02041) | 0.06219 (0.00004) | −4120.942 |
| WSELF | 0.74054 (0.00160) | 0.10734 (0.00056) | 0.44265 (0.04609) | 0.06150 (0.00068) | −4120.876 |
| MSELF | 0.73894 (0.00216) | 0.10677 (0.00527) | 0.38906 (0.12106) | 0.06081 (0.01121) | −4121.926 |
| PLF | 0.74296 (0.00161) | 0.10818 (0.00056) | 0.50920 (0.04090) | 0.06253 (0.00067) | −4121.245 |
| KLF | 0.74135 (0.00217) | 0.10762 (0.00525) | 0.46513 (0.10154) | 0.06184 (0.01109) | −**4120.797** |
| **Data** | **DS2** | | | | |
| **Bayesian Estimation** | | | | | |
| **Loss Function** | $\widehat{\lambda}$ ($r_{\widehat{\lambda}}$) | $\widehat{\theta}$ ($r_{\widehat{\theta}}$) | $\widehat{\alpha}$ ($r_{\widehat{\alpha}}$) | $\widehat{\omega}$ ($r_{\widehat{\omega}}$) | $\ell_{max}$ |
| SELF | 0.000275 ($1.889 \times 10^{-9}$) | 0.0366 ($3.6 \times 10^{-5}$) | 6.90240 (1.0208) | 0.1038 (0.0019) | −793.234 |
| WSELF | 0.000268 ($6.411 \times 10^{-6}$) | 0.0356 (0.0010) | 6.75207 (0.1503) | 0.0824 (0.0214) | −793.4759 |
| MSELF | 0.000262 (0.022395) | 0.0345 (0.0309) | 6.60050 (0.0224) | 0.0605 (0.2651) | −794.104 |
| PLF | 0.000278 ($6.821 \times 10^{-6}$) | 0.0371 (0.0009) | 6.97590 (0.1471) | 0.1128 (0.0179) | −**793.208** |
| KLF | 0.000271 (0.023714) | 0.0361 (0.0285) | 6.82680 (0.0221) | 0.0925 (0.2448) | −809.881 |

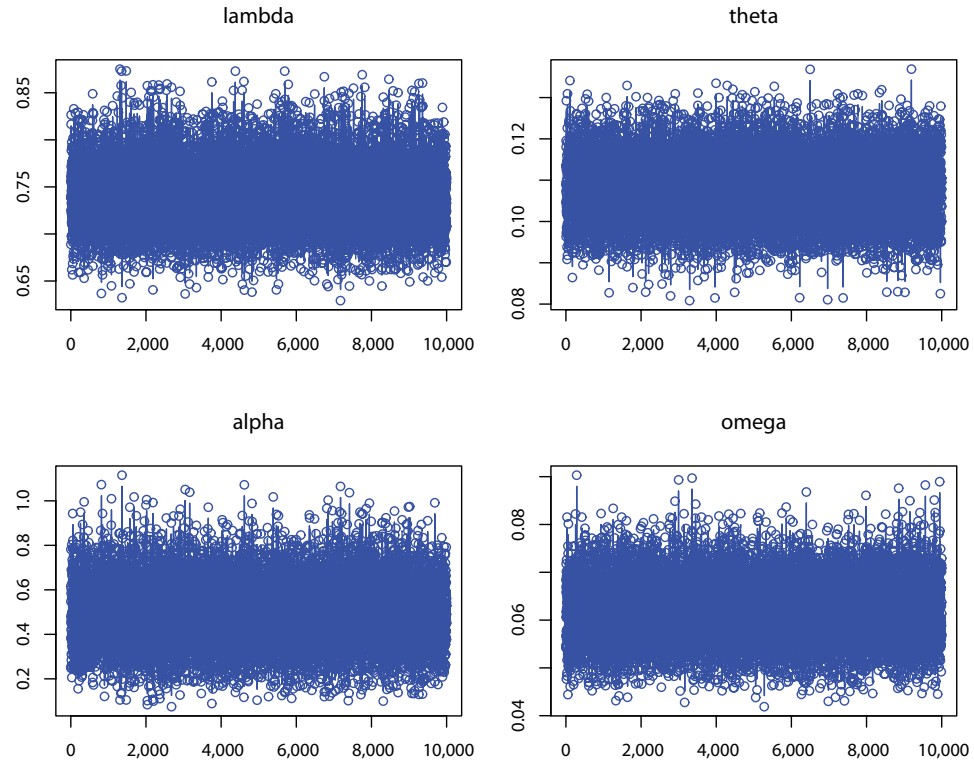

**Figure 6.** Plots of Bayesian analysis and performance of Gibbs sampling for DS1 dataset. Trace plots of each CWE distribution parameter.

**Table 5.** Credible and HPD intervals of parameters $\lambda$, $\theta$, $\alpha$ and $\omega$ for DS1 and DS2 datasets.

| Data | DS1 | |
|---|---|---|
| | **Credible Interval** | **HPD Interval** |
| $\lambda$ | (0.7184, 0.7645) | (0.6740, 0.8108) |
| $\theta$ | (0.1025, 0.1132) | (0.09299, 0.12300) |
| $\alpha$ | (0.3901, 0.5777) | (0.2336, 0.7923) |
| $\omega$ | (0.05767, 0.06658) | (0.04907, 0.07438) |
| **Data** | **DS2** | |
| | **Credible Interval** | **HPD Interval** |
| $\lambda$ | (0.00024, 0.00030) | ( 0.00019, 0.00035) |
| $\theta$ | (0.03269, 0.04018) | (0.02322, 0.04958) |
| $\alpha$ | (6.16500, 7.68300) | (5.04200, 8.76500) |
| $\omega$ | (0.07135, 0.12980) | (0.03289,0.19750) |

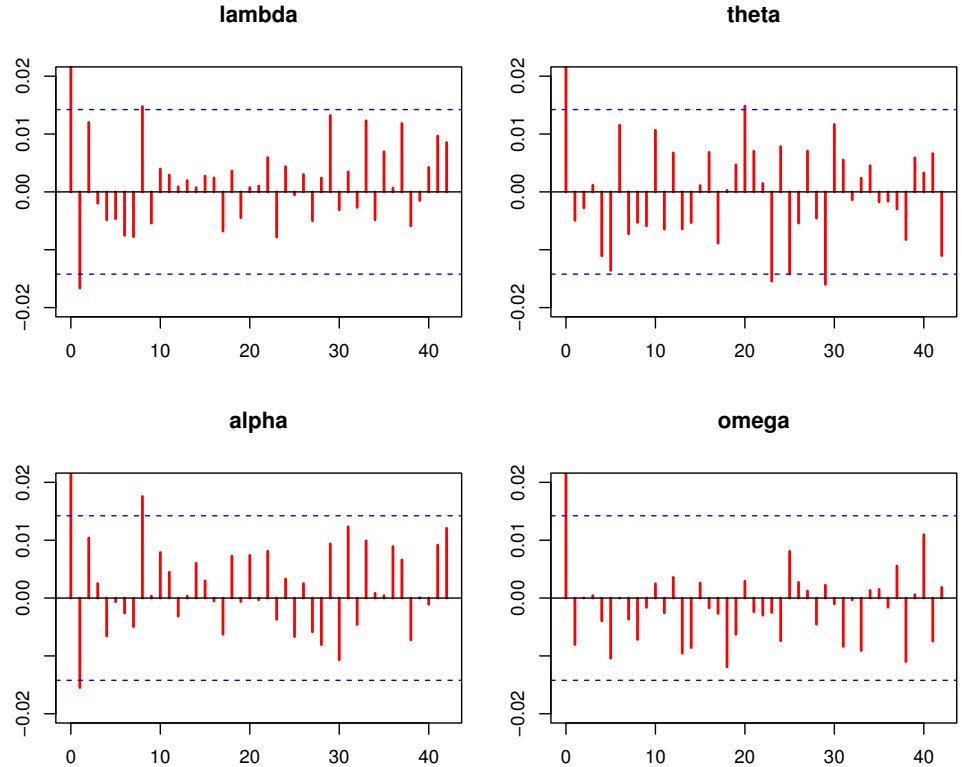

**Figure 7.** Plots of Bayesian analysis and performance of Gibbs sampling for DS1 dataset. Autocorrelation plots of each CWE distribution parameter.

In order to avoid repetition in evaluation of the MCMC procedure in Bayesian analysis, we just reported the Gelman–Rubin and Geweke–Raftery–Lewis diagnostics measures for checking the convergence based on data set DS1 in Table 6. For more details on these indexes see Lee et al. (2014). The Gelman–Rubin diagnostic is equal to 1 for parameters $\lambda$, $\theta$, $\alpha$ and $\omega$. Hence, the chains could be accepted, and this indicates the estimates come from a state space of the parameter, as depicted in Figure 9.

**Table 6.** Diagnostics using the Gelman-Rubin and Geweke-Raftery-Lewis methods for parameters $\alpha$, $\beta$ and $\lambda$ based on DS1 dataset.

| Parameter | Gelman-Rubin | Geweke ($Z_{0.025} = \pm 1.96$) | Raftery-Lewis |
|:---:|:---:|:---:|:---:|
| $\lambda$ | 1 | $-0.5880$ | 5.1 |
| $\theta$ | 1 | 0.3205 | 4.8 |
| $\alpha$ | 1 | 0.7607 | 5.01 |
| $\omega$ | 1 | 0.3679 | 4.632 |

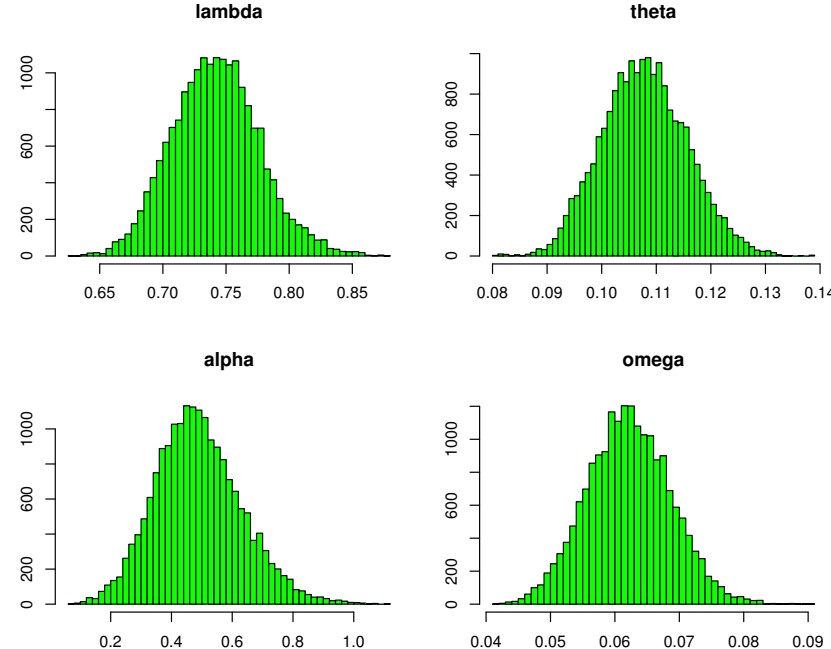

**Figure 8.** Plots of Bayesian analysis and performance of Gibbs sampling for DS1 dataset. Histogram plots of each CWE distribution parameter.

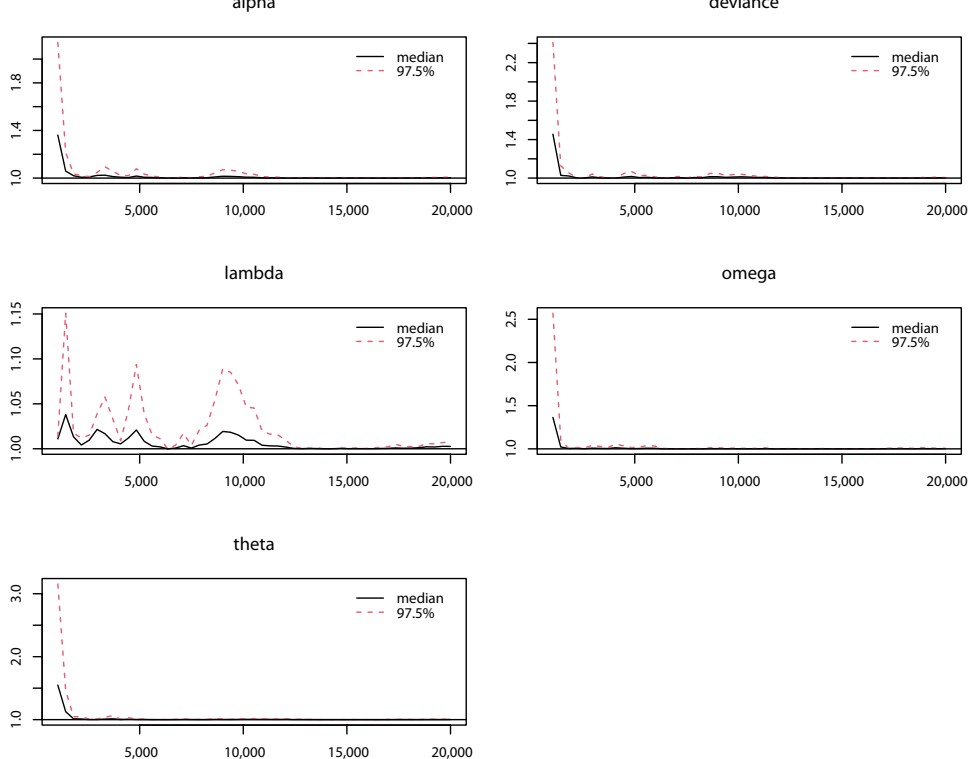

**Figure 9.** Gelman plot diagnostic for each CWE distribution parameter based on DS1 dataset.

From Table 6, Geweke–Raftery–Lewis test statistics for parameters $\lambda$, $\theta$, $\alpha$ and $\omega$ are $-0.588$, $0.320$, $0.761$ and $0.368$, respectively. Therefore, also in this case, the chain is acceptable, as shown in Figures 10 and 11. Moreover, the reported diagnostics statistics for parameters $\alpha$, $\beta$ and $\lambda$ based on the Geweke–Raftery–Lewis measure don't show significant correlations between estimates. Hence, the estimated values have good mixing.

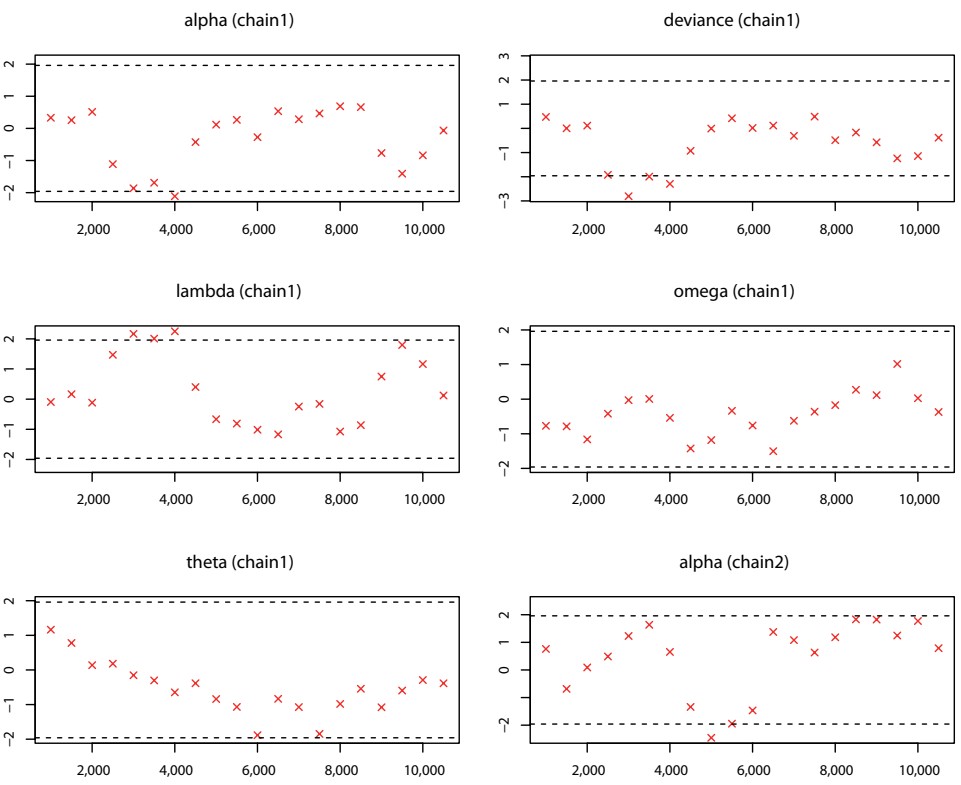

**Figure 10.** Geweke plot diagnostic (chain1) for each CWE distribution parameter based on DS1 dataset.

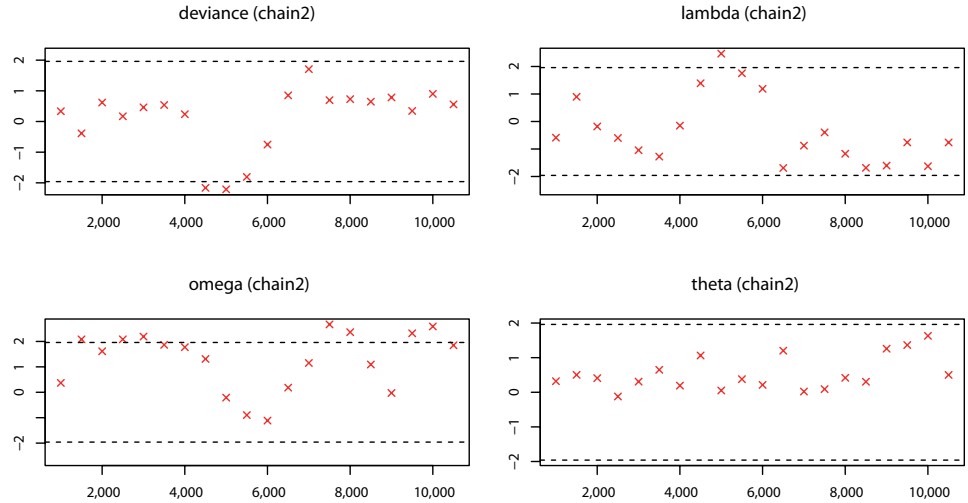

**Figure 11.** Geweke plot diagnostic (chain2) for each CWE distribution parameter based on DS1 dataset.

## 8. Conclusions

This paper extended the WE distribution to a richer family, the CWE distribution, to deal with data displaying large and positive skewness as well as a wide right tail. This four-parameter model is a mixture of two WE distributions in which one has an enhanced

scale and hence a thicker tail to capture extreme losses. EM and Bayesian computational techniques were used to estimate parameters. The effectiveness and efficiency of the EM algorithm were evaluated by conducting one simulation study. By analyzing two real insurance claims datasets, we found that the CWE distribution outperformed the CE distribution in terms of model fit. The result show that both EM and Bayesian approaches are appropriate tools to estimate the model parameters. In addition, it is possible to consider proposed distribution to fit lifetimes, and how the suggested algorithms will be adjusted in case of truncated or censored data. Another application could be done in actuarial science context; specifically, how CWE distribution could be employed to calculate the VaR and TVaR (Bargès et al. 2009).

**Author Contributions:** Conceptualization, A.M. and O.K.; methodology, A.M. and O.K.; software, A.M. and O.K.; validation, A.M., O.K. and J.E.C.-R.; formal analysis, O.K.; investigation, A.M., O.K. and J.E.C.-R.; resources, J.E.C.-R.; data curation, A.M.; writing—original draft preparation, A.M., O.K. and J.E.C.-R.; writing–review and editing, A.M., O.K. and J.E.C.-R.; visualization, A.M. and O.K.; supervision, J.E.C.-R.; project administration, O.K.; funding acquisition, J.E.C.-R. All authors have read and agreed to the published version of the manuscript.

**Funding:** This research was fully supported by FONDECYT (Chile) grant No. 11190116.

**Data Availability Statement:** The datasets analyzed during the current study are available from the corresponding author on reasonable request.

**Acknowledgments:** The authors also thank the editor and two anonymous referees for their helpful comments and suggestions. All R and OpenBUGS codes used in this paper are available upon request from the corresponding author. The datasets analyzed during the current study are available from the corresponding author on reasonable request.

**Conflicts of Interest:** The authors declare that there is no conflict of interest in the publication of this paper.

## Appendix A. R Code to Fit the CWE Distribution Using EM-Algorithm

```
EM.CWE <- function(y, om, al, la, th, iter.max = 500, tol=10^-6){
f.CWE <- function(y,om,al,la,th)
(1-om)*(al+1)/al*la*exp(-la*y)*(1-exp(-la*al*y))+om*(al+1)/al*th*la
*exp(-th*la*y)*(1-exp(-th*la*al*y))
n <- length(y); LL <- 1 ; dif <- 1 ; count <- 1
while ((dif > tol) & (count <= iter.max)) {
# E steps
gam <-  om*(al+1)/al*th*la*exp(-th*la*y)*(1-exp(-th*la*al*y))/
f.CWE(y,om,al,la,th)
ta1 <- (1-gam)*(1/(la*al*y)-1/(exp(la*al*y)-1) )
ta2 <- gam*(1/(th*la*al*y)-1/(exp(th*la*al*y)-1) )
# M steps
om <- sum(gam)/n
al <- n/(la*sum(ta1*y+th*ta2*y))-1
la <- 2*n/sum((1-gam)*y+al*ta1*y+th*gam*y+al*th*ta2*y)
th <- (2*sum(gam))/(la*sum(gam*y+al*ta2*y))
LL.new <- sum(log(f.CWE(y,om,al,la,th)))
count <- count +1
dif <- abs(LL.new/LL-1)
}
print.foo <- function(x) print(x[1:8])
aic <- -2 * LL.new + 2 * 4
bic <- -2 * LL.new + log(n) * 4
Ret <-list(omega=om, alpha=al, lambda=la, theta=th, loglik=LL.new,
AIC=aic, BIC=bic, iter=count, out.prob=gam)
class(Ret) <- "foo"
return(Ret)
}
```

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
