# Peer review of "On the Contaminated Weighted Exponential Distribution: Applications to Modeling Insurance Claim Data"

_jrfm, doi:10.3390/jrfm15110500_

Round 1
Reviewer 1 Report
Article Review: On the contaminated weighted exponential distribution: Applications to modeling insurance claim data
This paper presents two strategies (EM and Bayesian) to estimate the parameters of the Contaminated Weighted Exponential family (CWE) that offers advantages when the distribution of losses presents heavy tails with some outliers.
Firs of all, I want to congratulate you on the work, I think it can be an interesting contribution to treat data with distributions of heavy tails. However, I have several doubts. I fail to understand which of the two proposed estimation strategies is better. What advantages does the Bayesian strategy have over EM? I particularly believe that many, however, are not valued in this article.
In section 5. Simulation study: recovery of the true underlying parameters, it would be good to show the posterior density functions of the parameters, the errors’ distribution, or a plot in which the estimation of the parameters and its credible intervals are shown, since simulations are carried out with different values of n in the sample.
On the other hand, in section 7. Bayesian numerical results hardly show any novelty, since only the results related to the Markov chain obtained for each parameter when applying the Gibbs sampler are presented, they are optimal considering a good choice of hyperparameters.
In addition, it would be interesting to make a comparison between both estimation strategies, showing which is more accurate. In relation to the conclusions, they must be more extensive. Finally, the notation of the paper should be reviewed, as there are several errors of this type mixing notations.
Author Response
Please see attached response letter.

Reviewer 2 Report
The manuscript introduces an extension of the weighted exponential (WE) distribution (Gupta and Kundu 2009) by using a mixture of the WE. Overall the paper is well-written and the results are well presented, however the authors are asked to provide some clarifications before considering this paper for publication.
Major comments:
1) The proposed model is for fitting insurance data. While the statistical analysis provides enough insight on how the estimation method suggested by the authors is giving a good fit to insurance data, I couldn't see any discussion of the properties of this new distribution. It will be good to derive the typical statistical measures and study how the added parameters from WE to CWE,(i.e. Theta and Omega) contribute to these properties. For example, what impact Theta and Omega have on the shape of the tails, the skewness, and the kurtosis?
2) In actuarial science, it is important to have a good fit of the right tail. In such a context, it will be good to calculate the VaR, Stop-Loss, and TVaR using the different considered distributions and analyze how the CWE performs compared to the other distributions.
3) While the VaR is not going to have a closed-form using the CWE, the authors can easily derives expressions for the Stop-Loss premiums, and TVaR.
4) I understand that the main scope of the paper is fitting the CWE to insurance claims data, but is it possible to consider this distribution to fit lifetimes, and how the suggested algorithms will be adjusted in case of truncated or censored data. The authors may consider this in their future research.
I am looking forward to reviewing the updated version of this paper.
Author Response
Please see attached response letter.

Reviewer 3 Report
Questions and inquiries in the paper must be answered and answered in a separate report:
A paragraph should be added on the organizational part of the paper.
What is the motivation for creating a new model using mixture system?
In line 148-150, How is this function drew where this is a complex function?
Figure 1, should be changed as bold lines.
The EM code must be attached to the appendix, as it is not clear what was done in it.
How were the hyper-parameters values chosen?
Why choose these hyper-parameter values are set as ai = qi = li = wi = 0.0001?
Why use OpenBUGS?
The simulation should be repeated when the sample size is as follows 30, 70, 100, 200.
generate 500 synthetic Monte Carlo samples, this isn't sufficient to make convergence
Figure 3 Reveals a problem with MCMC
In Table 4, L_max !!!!???? what this and how calculate it in Bayesian?
Author Response
Please see attached response letter.

Round 2
Reviewer 1 Report
First of all, I would like to thank the authors for taking into account the suggestions made and responding to the issues raised in the first revision.
After a second review, I believe that the paper can be accepted for publication in this journal, as the errors have been corrected and some of the issues raised have been added. However, I still believe that the conclusions could be improved.
Nevertheless, I would like to urge the authors to consider a Bayesian simulation study, of the model proposed in this paper, in the future.
Reviewer 3 Report
Metropolis-Hastings algorithm can be improved by see Accelerated life tests for modified kies exponential lifetime distribution: Binomial removal, transformers turn insulation application and numerical results
Credibility interval can be improved by see Bayesian analysis in partially accelerated life tests for weighted lomax distribution
The theory of Weighted distribution can be improved by see Weighted Power Lomax Distribution and its Length Biased Version: Properties and Estimation Based on Censored Samples and Bayesian Analysis in Partially Accelerated Life Tests for Weighted Lomax Distribution